# MMQA: Evaluating LLMs with Multi-Table Multi-Hop Complex Questions

**Jian Wu**[1*]  **Linyi Yang**[2*]  **Dongyuan Li**[4*]  **Yuliang Ji**[5]  **Manabu Okumura**[1]  **Yue Zhang**[3†]

[1]Institute of Science Tokyo [2]University College London [3]School of Engineering, Westlake Univeristy
[4]The University of Tokyo [5]Nanjing University of Science and Technology.

## Abstract

While large language models (LLMs) have made strides in understanding tabular data, current tabular evaluation benchmarks, such as WikiTableQuestions and WikiSQL, are focus on single-table scenarios, which cannot necessarily reflect the complexity of real-world applications. To bridge this gap, we present a **M**ulti-table and **M**ulti-hop **Q**uestion **A**nswering (MMQA) dataset to assess LLMs' understanding and reasoning capabilities in handling multi-table tasks. The MMQA dataset demands that models perform multiple inferences by drawing evidence from various tables, which are designed to be connected and require models to identify and utilize relationships such as foreign and primary keys. Then, we introduce a comprehensive evaluation framework that tailors to assess LLMs' capabilities in several aspects including Multi-Table Retrieval, Text-to-SQL Generation, Multi-Table QA, Primary Key Selection, and Foreign Key Selection. Finally, we propose a novel multi-table retrieval method that achieves state-of-the-art (SOTA) performance on the MMQA dataset compared to several strong baselines. Our experiment results reveal that, compared with human performance, both open-source and commercial LLMs leave significant performance room for improvements in multi-table understanding and reasoning tasks. We believe that the MMQA benchmark will enhance and facilitate LLMs' multi-table capabilities in real-world scenarios. The Whole MMQA data are available at `https://anonymous.4open.science/r/MMQA-34B1`

## 1 Introduction

*Table* is one of the fundamental structured data types in real-world scenarios, its widespread application includes relational databases and spreadsheet forms (Raffel et al., 2019). Recent studies have shown the strong capabilities of LLMs on table-related tasks (Zhu et al., 2021; Zhao et al., 2023; Hegselmann et al., 2022; Li et al., 2023; Zhang et al., 2024b; Lu et al., 2024). However, LLMs' multi-table understanding and reasoning performance remain relatively less explored. Previous table-related studies such as Table-QA (Chen et al., 2020b; Zhu et al., 2021; Pasupat & Liang, 2015; Zhong et al., 2017; Yu et al., 2018a; Cheng et al., 2021; Katsis et al., 2021; Nan et al., 2022; Jauhar et al., 2016; Li et al., 2021; Chen et al., 2020a), Table Fact Verification (Chen et al., 2019; Günther et al., 2021), Table-to-text Generation (Moosavi et al., 2021; Suadaa et al., 2021; Lebret et al., 2016), and Column type & Relation classification (Iida et al., 2021; Deng et al., 2020) all focus on single-table tasks. However, in real-world scenarios, operations such as join, union, intersection, and foreign key identification are frequently used in multi-table reasoning, but there have been very few benchmarks for their comprehensive evaluation (Pal et al., 2023; Zhang et al., 2024a).

To fill in this gap, we build a **M**ulti-table and **M**ulti-hop **Q**uestion **A**nswering evaluation datasets (MMQA), aiming to evaluate LLMs' understanding and reasoning on multi-table data. For a comprehensive understanding of LLMs' performance on multi-table table tasks, we propose an evaluation framework over different aspects (Multi-table Retrieval, Text-to-SQL Generation, Multi-table

---

*Equal contribution. Jian Wu did this work during his internship at Westlake University
†Corresponding author.

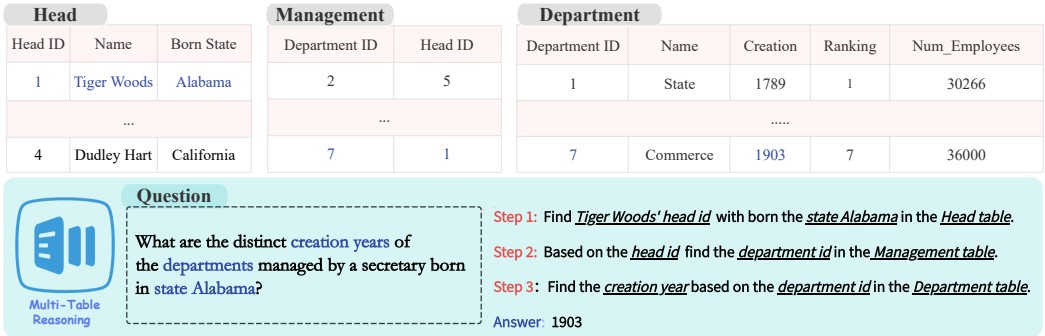

Figure 1: An example from our MMQA benchmark. Typical challenges in MMQA are: 1) Determine the reasoning order; 2) Identify the primary keys and foreign keys; 3) Retrieve pieces of evidence from different tables for multi-hop reasoning. The words in deep blue in tables are evidence for reasoning the answer.

Question Answering, Foreign Key Selection, and Primary Key Selection), jointly considering coarse-grained and fine-grained information of tables, at the table, column, and cell level. Figure 1 depicts an example from our MMQA, a question involving three tables. The question is *"What are the distinct creation years of the departments managed by a secretary born in state Alabama?"* and the three input tables are *"Head"*, *"Management"* and *"Department"*. The words in blue are the keywords for reasoning across tables. LLMs are expected to first understand the question as well as the input tables to determine the reasoning order. Starting reasoning from a specific table and column, and then locating the key columns for jumping to another table. For instance, in the **Head** table, the columns are *Head ID*, *Name*, and *Born State*, where the column *Head ID* is both the foreign key and the primary key of the table, which helps LLMs to jump to the table **Management**. In the **Head** table, we can determine the *head ID* is *"1"* based on the state *"Alabama"*. Then, from *Head ID* to *Department ID* in the **Management** table, LLMs could determine the number *"7"* department. Finally, based on the department *"7"* in **Department** table, we can find the candidate answer *"1903"*. Consequently, in this example, LLMs need to reason intra-table and inter-table at both column level and cell level, which is much more complex than single-table reasoning. The differences between multi-table qa and single-table qa are reminiscent of those between Multi-hop QA (Yang et al., 2018) and single-hop QA (Rajpurkar et al., 2016).

We conduct extensive experiments to assess the multi-table and multi-hop understanding and reasoning abilities of the LLMs on our MMQA dataset. The results demonstrate the superiority of human performance over current SOTA LLMs, shedding light on the challenges encountered by existing models in performing multi-table tasks. Based on that, we propose a novel multi-table retrieval method, named **MTR**, which incorporates the Question Decomposition module to decompose a multi-hop question into a series of sub-questions and converts multi-table retrieval task into several rounds of single-table retrieval task. In each round, MTR jointly considers question-table relevance and table-table relevance for single-table retrieval. To the best of our knowledge, our research is the first to introduce a multi-table and multi-hop QA benchmark, evaluating LLMs' multi-table complex reasoning abilities. On the MMQA dataset, MTR shows an advantage in achieving the best results over a range of the previous strong baselines.

## 2 RELATED WORK

**Single-Table QA.** Table Question Answer (TQA) involves retrieving answers from one or several table cells from a given table, such as WikiTableQuestions (Pasupat & Liang, 2015), WikiSQL (Zhong et al., 2017), SPIDER (Yu et al., 2018a), TABFACT (Chen et al., 2019). However, these datasets mainly focus on reasoning on tables and ignore important knowledge stored in the textual corpus. Consequently, QA covering both tabular and textual knowledge has gained increasing interest. Chen et al. (2020b) pioneered a passage-table QA benchmark, HybridQA, with Wikipedia tables linked to relevant free-form text passages (e.g., Wikipedia entity-definition pages). The OTT-QA (Chen et al., 2020a) benchmark extended HybridQA to the open domain setting, where a system needs

to first retrieve a relevant set of tables and passages before trying to answer questions. Moreover, the links from the table and the passage are not explicitly provided. FinQA (Chen et al., 2021) and AIT-QA (Katsis et al., 2021) are predominantly target financial and airline tables, suggesting complex reasoning challenges that require models not only to interpret but also to compute and extract nuanced information precisely. TableBench (Wu et al., 2024b), a comprehensive and complex tabular benchmark, including 18 fields within four main categories to evaluate the TQA capabilities of LLMs. Despite the significant advances made by LLMs in TQA (Li et al., 2022; Singha et al., 2023; Li et al., 2023), there is still a critical need for benchmarks that reflect the multi-table reasoning complexity encountered in real-world scenarios. Our work differs from this line of work, incorporating real-world complexities into its evaluation scenarios.

**LLM for Table Reasoning.** Despite the remarkable performance of LLMs in textual reasoning, their reasoning capabilities on tabular tasks are still limited. Zhu et al. (2021) proposes a TAT-LLM for reasoning over a hybrid of tabular and textual data, on FinQA (Chen et al., 2021), TAT-QA (Zhu et al., 2021) and TAT-DQA Zhu et al. (2022) benchmarks. Cheng et al. (2022) and Chen (2022) focus on utilizing LLMs to reason over single-table data with a zero-shot setting. TableLLM (Zhang et al., 2024b) and TableLlama (Zhang et al., 2023) are two table-related LLMs, pre-trained and evaluated on single-table datasets. TAT-LLM (Zhu et al., 2024) tackles the question-answering task (QA) by proposing a step-wise pipeline including Extractor, Reasoner, and Executor to assist LLMs in better performing discrete reasoning over a hybrid of tabular and textual data. Ye et al. (2023) harnesses the multi-step reasoning capabilities of LLMs to first decompose complex questions into sub-questions by generating intermediate SQL queries for tabular data. TAP4LLM (Sui et al., 2023) is a versatile pre-processing toolbox to generate table prompts to enhance the complex reasoning ability of LLMs over tabular data. However, all LLM-based table reasoning methods focus on single-table reasoning. The multi-structure understanding and reasoning problem remains under exploration.

**Multi-Table Tasks.** Pal et al. (2023) pioneer a multi-table pre-training task of answering questions over multiple tables and targets to generate sub-tables from input multi-tables. Liu et al. (2023) furtherly proposes a document-level summarization dataset that jointly considers textual information as well as multi-table content in documents. However, the two methods focus on the sub-table level, which is coarse-grained. In contrast, consider the multi-hop and multi-table tasks. Chen et al. (2024) propose a multi-table retrieval method that considers the relevance between column and sub-questions, without alignment of whole table headers. Our work is different from previous works in two main aspects: 1) a comprehensive multi-table understanding and reasoning evaluation; and 2) evaluation on different granularities, i.e., table, column, and cell levels.

## 3 MMQA

As shown in Figure 2, the evaluation framework has two main steps: 1) Multi-table Retrieval; and 2) Multi-table Evaluation. The evaluation tasks consist of four categories: Multi-table retrieval, Text-to-SQL Generation, Multi-table Question Answering, and Key Selection (primary key and foreign key).

### 3.1 MMQA CONSTRUCTION

This section details the dataset construction process, including data annotation and quality verification, and reasoning steps compared with the previous Table QA benchmarks.

**Data Collection and Annotation** We develop the MMQA benchmark over Spider database (Yu et al., 2018a), which is a cross-domain complex semantic parsing dataset for Text-to-SQL Generation. Spider consists of 5,693 SQL queries and more than 200 databases of multiple tables covering 138 different domains. We randomly select a total of 5,000 samples from Spider and each sample contains two or three tables.

**Question Answer Annotation** Then, we follow the paradigm of Pal et al. (2023) to synthesize multi-table SQL queries through the 45 manually crafted templates over the Spider database and hand-crafted rules for operation types. After getting the SQL queries, we prompt the tables and SQL queries into LLMs such as GPT-4-turbo to generate natural language questions. Finally, we asked two human experts with computer science backgrounds to annotate the foreign keys and primary keys of each table. The primary key of one table is a column or a constraint that uniquely identifies each record in the table. The foreign key is a column or combination of columns that is used to establish

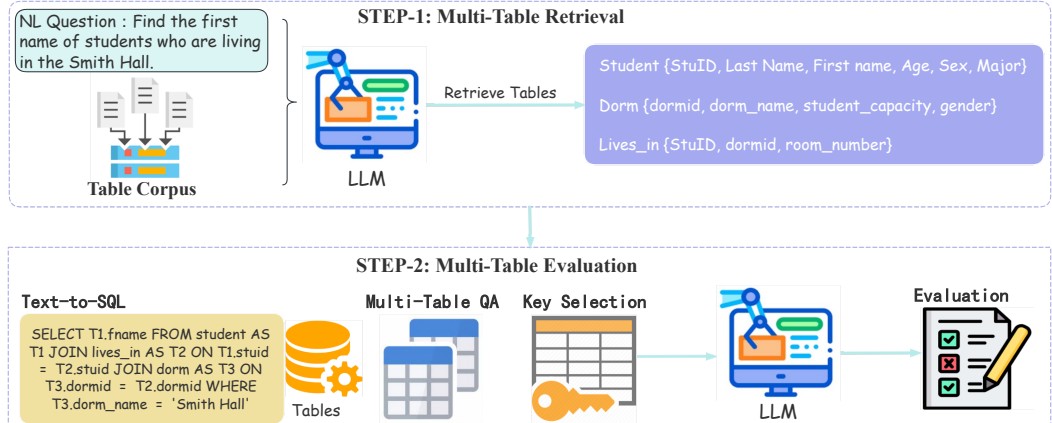

Figure 2: The framework of our multi-table evaluation, LLMs are firstly required to retrieve tables from a given table corpus. Then, we evaluate LLMs' reasoning and understanding abilities on MMQA. The multi-table evaluation involves Text-to-SQL Generation, Multi-Table Question Answering, Primary Key Selection, and Foreign Key Selection.

and enforce a link between the data in two tables to control the data that can be stored in the foreign key table. For answer annotation, two human experts are required to give answers based on the tables and generated questions.

**Quality Verification** After obtaining the annotation results, in cases of discrepancies, a third expert was invited to review the annotations of the two experts, serving as guidelines to improve consistency, and the final results were determined by majority voting. To ensure the quality of the annotated data, we discard the questions whose correct answers could not be extracted from the given table or have grammar issues. The inter-agreement is the average score computed based on the three experts' check results. We compute the results of experts as human performance, compared with LLMs. We list the properties of our benchmark in Table 1.

**Question Categories** Drawing from real-world scenarios and user demands for multi-table data, we design four primary question categories: *Numerical* (numeric operation, sum, average, etc.),

Table 1: Summary of statistics of MMQA and inter-human agreement.

| Properties | 2 table | 3 table |
|---|---|---|
| Total Tables | 2591 | 721 |
| Avg rows per table | 1833.31 | 1369.01 |
| Avg columns per table | 6.04 | 4.78 |
| Avg foreign keys per table | 2.81 | 1.95 |
| Avg primary keys per table | 3.35 | 2.41 |
| Inter-human agreement | 86% | 82% |
| Question Length | 77.11 | 85.38 |
| **Question Category** | | |
| Numerical | 889 | 289 |
| List | 214 | 44 |
| Count | 200 | 42 |
| Select | 1289 | 346 |

*List* (list operation that showcases all answers meet the conditions), *Count* (count the number of answers that meet the conditions), and *Select* (select a specific answer that meets the conditions). We illustrate several examples of different categories of questions in Table 3.

**Reasoning Steps** We compare the data complexities of different datasets by calculating the number of reasoning steps required to solve the multi-hop questions. Figure 3 demonstrates that the average reasoning step of MMQA is significantly higher than that of existing datasets.

Finally, after obtaining MMQA, a comprehensive and complex benchmark consisting of 3,312 tables in total with the corresponding natural language question, SQL query, gold answer, foreign key, and primary key annotation.

## 3.2 MULTI TABLE RETRIEVAL (MTR)

Different from the single-table retrieval task, a multi-table retrieval task over a couple of tables can be defined as follows. Specifically, given a question $Q$, a table corpus $C = \{T_i\}_{i=1}^{M}$, retrieve

Table 2: Differences between our MMQA with previous table QA benchmarks. We here abbreviate the natural language as 'NL'. Our benchmark can be applied to evaluate LLM's multi-table understanding and reasoning abilities more comprehensively.

| Benchmarks | Question format | Data size | Data source | Task | Multi-table |
|---|---|---|---|---|---|
| WTQ (Pasupat & Liang, 2015) | NL question | 20,000 table-question pairs | Wikipedia | Single-table QA | ✗ |
| WikiSQL (Zhong et al., 2017) | SQL query | 24241 tables | Wikipedia | Single-table QA | ✗ |
| HybridQA (Chen et al., 2020b) | NL question | 70k table-question pairs | Wikipedia | Table-text QA | ✗ |
| SQA (Iyyer et al., 2017) | NL question | 6,066 unique questions | Wikipedia | Single-table QA | ✗ |
| FeTaQA (Nan et al., 2022) | NL question | 10,330 tables | Wikipedia | Single-table | ✗ |
| Spider (Yu et al., 2018b) | NL question & SQL query | 8000 questions and SQL query pairs | Crowdsourcing | Text-to-SQL | ✗ |
| BIRD (Li et al., 2024) | NL question & SQL query | 12,751 questions and SQL query pairs | Kaggle | Text-to-SQL | ✗ |
| SPINACH (Liu et al., 2024) | NL question & SQL query | 320 questions and SQL query pairs | Crowdsourcing | Text-to-SQL | ✗ |
| Tablebench (Wu et al., 2024b) | NL | 3681 tables | WTQ/SQA/FeTaQA /FinQA/AIT-QA | Single-table | ✗ |
| **MMQA**(Ours) | NL question & SQL query | 3,312 tables | Wikipedia | Multi-table retrieval, Text-to-SQL, Multi-table QA Primary Key & Foreign Key Selection | ✓ |

Table 3: Examples of MMQA question types and tables. We emphasize keywords for the related table columns.

| Table Type | Question Type | Multi-hop Question |
|---|---|---|
| 2 table | Numerical | What are the *ids of all stations* that have a *latitude* above 37.4 and have never had less than 7 *bikes* available? |
| | List | List the *customers' first and last name* of 10 least expensive *invoices*. |
| | Count | How many *departments* are led by *heads* who are not mentioned? |
| | Select | What are the *ids of the courses* that are *registered* or *attended* by the *student* whose id is 121? |
| 3 table | Numerical | What is the *salary* and *name* of the *employee* who has the most number of *certificates* on *aircraft* with *distance* of more than 5000? |
| | List | Find the *cell mobile number* of the *candidates* whose *assessment code* is Fail? |
| | Count | For each *course id*, how many *students* are registered and what are the *course names* ? |
| | Select | What are the distinct *creation years* of the *departments* managed by a secretary born in *state 'Alabama'?* |

a table $T_i$ from $C$ that contains the answer of $Q$. Single-table retrieval task is to select the most question-related table from $C$. However, in the multi-table retrieval task, our problem is to retrieve a list of question-related tables, which can be joint reasoning with the connection of foreign keys. Here, we denote the retrieved tables as $R(t) = \{t_1, t_2, ...\}$, where $t_i \in C$ and $t_i$ must be joinable with another $t_j \in R(t)$. For example, in Figure 1, the table `Head` is joinable with table `Management` because they have the same column `Head ID`. Consequently, to identify the most related tables over a corpus of tables, we need to consider two aspects: retrieving the question-related tables, and retrieving the table-related tables. Inspired by the previous multi-hop question decomposition work (Wu et al., 2024a), which generatively decomposes the multi-hop questions for enhancing question-related evidence retrieval performance.

We propose a novel multi-table retrieval method (MTR) that iteratively retrieves question-related and table-related tables. Given a multi-hop question $Q$, we first use the GPT-4-turbo as the question decomposer and feed the multi-hop question into the decomposer directly by a set of prompts (Appendix B) with a one-shot setting and get a series of sub-questions $q_1, q_2, ...q_n$. Then, we fine-tune TableLlama-7b (Zhang et al., 2023), as well as SGPT-5.8B (Muennighoff, 2022), as the single-table retrieval model on the off-the-shelf sing-table QA datasets Chen et al. (2019; 2021); Katsis et al. (2021); Zhu et al. (2021).

We treat the multi-table retrieval task into several rounds of sing-table retrieval tasks. For $n$ decomposed sub-questions, we iterate $n$ rounds to retrieve the top K (K=2,5,10) sub-question-related tables from the table corpus. In the first round, we only consider the question-table relevance score, and each retrieved table is assigned a table-relevance score. Then we rank the scores to get the top K

---

**Algorithm 1** Multi-Table Retrieval

---

**Initialize:**
**Input:** Multi-hop Question $Q$, **LLM:** GPT-4-turbo.
**Ouput:** Retrieved Tables
**First Round:** $\gamma \leftarrow \gamma + \alpha(q_0, table_j^0)$     ▷ only compute question-relevance scores in 1st round
**for** $i \in 1$ **to** $n$ **do**
    **for** $j \in 0$ **to** $M$ **do**
        **for** $k \in 0$ **to** $M$ **do**            ▷ Compute Relevance Scores
           $\gamma \leftarrow \gamma + \alpha(q_i, table_j^i) \cdot \beta(table_k^{i-1}, table_j^i)$
        **end for**
    **end for**
**end for**
**for** $i \in$ **ArgSort**$(\gamma$, descending=True) **do**
    $table_i \leftarrow \max(\gamma, table_i)$            ▷ Select top K relevant tables
**end for**
**Return** tables

---

tables. From the second round to the end, the previously retrieved tables are treated as seeds. Where we rank the top K tables based on the question relevance score as well as the table relevance score. The question relevance score is computed with a single-table retrieval model, and the table relevance score is computed based on the overlap of table columns. If the retrieved tables overlap columns with tables retrieved in the previous round, then assign a score of 1 and 0. When 0 is assigned, stop iterations. In each round, the score is the product of the question relevance score and table relevance score. After all the sub-questions have been used to retrieve the tables, we sum all the scores:

$$\gamma = \sum_{i \leq n, j \leq M, k \leq M} \alpha(q_i, table_j^i) \cdot \beta(table_k^{i-1}, table_j^i) \tag{1}$$

where $\alpha(q_i, table_j^i)$ is the relevance score, given by the single table retrieval model, between $i$th sub-question with $j$th retrieved table in round $i$. $\beta(table_k^{i-1}, table_j^i)$ is the table relevance score between the $k$-th retrieved table in $i-1$ round and $j$-th retrieved table in round $i$. The implementation details are illustrated in Appendix C.

## 3.3 SUBTASKS

The multi-table evaluation is referred to as finding answers to complex questions that require reasoning multiple times from given tables. We employ a multi-faceted metric set for Multi-Table Retrieval, Multi-Table Question Answering, Text-to-SQL, and Key Selection.

**Multi-Table Retrieval.** We evaluate the LLMs' performance on the Multi-Table retrieval task on MMQA whose questions require reasoning over multiple tables to answer. The goal is to answer the following question: To what extent can LLMs retrieval multi-table that jointly considers question-table relevance and table-table relevance? We build a table pool that includes

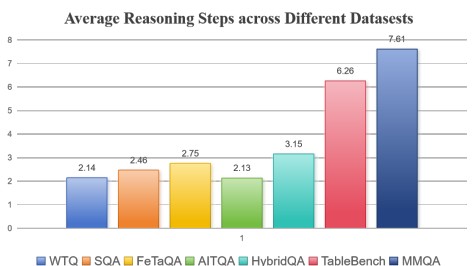

Figure 3: Data complexity comparison with existing datasets in reasoning steps.

all tables of MMQA and input multi-hop questions into LLMs to retrieve all question-related tables from the pool. This task is fundamental because in the real-world scenario, the subsequent steps of reasoning and answering performance are based on the quality of the retrieved tables.

**Text-to-SQL.** Following the text generation evaluation task, we utilize Rouge-1, Rouge-L (Lin, 2004), and BLEU (Papineni et al., 2002) to evaluate the LLM-generated SQL query quality against

Table 4: The main experiment results on Multi-Table Retrieval, our MTR outperforms all previous strong baselines.

| | Top-2 | | | | | | Top-5 | | | | | | Top-10 | | | | | |
| | 2-table | | | 3-table | | | 2-table | | | 3-table | | | 2-table | | | 3-table | | |
| | P | R | F1 | P | R | F1 | P | R | F1 | P | R | F1 | P | R | F1 | P | R | F1 |
| --- | --- | --- | --- | --- | --- | --- | --- | --- | --- | --- | --- | --- | --- | --- | --- | --- | --- | --- |
| *Baselines* | | | | | | | | | | | | | | | | | | |
| BM25 | 6.2 | 4.3 | 5.1 | 4.4 | 5.2 | 4.8 | 9.8 | 8.7 | 9.2 | 6.3 | 7.5 | 6.8 | 10.9 | 9.2 | 9.9 | 8.6 | 8.2 | 8.4 |
| tfidf | 4.6 | 5.1 | 4.8 | 4.5 | 5.1 | 4.8 | 10.6 | 9.3 | 9.9 | 8.5 | 9.2 | 8.8 | 11.4 | 10.8 | 11.1 | 9.3 | 8.9 | 9.1 |
| DTR (Herzig et al., 2021) | 31.4 | 35.4 | 33.3 | 29.9 | 30.9 | 30.4 | 34.0 | 35.8 | 34.9 | 33.2 | 32.9 | 33.0 | 39.3 | 37.2 | 38.2 | 38.5 | 35.4 | 36.9 |
| SGPT-125M (Muennighoff, 2022) | 39.5 | 41.7 | 40.6 | 37.4 | 39.1 | 38.2 | 40.4 | 42.2 | 41.3 | 38.6 | 39.4 | 39.0 | 41.7 | 42.5 | 42.1 | 40.2 | 40.6 | 40.4 |
| SGPT-5.8B (Muennighoff, 2022) | 45.7 | 48.1 | 46.9 | 44.2 | 45.3 | 44.7 | 45.1 | 44.9 | 46.9 | 43.9 | 45.3 | 44.6 | 47.7 | 48.8 | 48.2 | 46.2 | 47.3 | 46.7 |
| TableLlama-7b (Zhang et al., 2023) | 56.7 | 58.2 | 57.4 | 53.6 | 52.8 | 53.2 | 59.2 | 63.1 | 60.1 | 57.7 | 58.1 | 57.9 | 60.8 | 64.2 | 62.5 | 59.6 | 59.3 | 59.4 |
| *Our methods —MTR* | | | | | | | | | | | | | | | | | | |
| MTR (SGPT-5.8B) | 58.1 | 53.9 | 55.9 | 51.4 | 49.3 | 50.3 | 62.3 | 59.5 | 60.9 | 60.2 | 64.7 | 62.4 | 61.7 | 65.6 | 63.6 | 61.8 | 63.5 | 62.6 |
| MTR (TableLlama-7b) | 72.3 | 64.7 | 68.3 | 69.5 | 66.2 | 67.8 | 74.3 | 71.8 | 73.0 | 72.9 | 70.6 | 71.7 | 74.5 | 73.3 | 74.9 | 73.6 | 71.9 | 72.7 |
| w/o QD | 65.3 | 62.3 | 63.8 | 64.7 | 63.6 | 64.1 | 70.2 | 68.3 | 74.6 | 68.6 | 67.5 | 68.0 | 70.8 | 68.5 | 69.6 | 69.5 | 67.7 | 68.6 |

ground truth. Different from single-table QA reasoning tasks, the multi-table SQL query is much more complex with more operations.

**Multi-Table Question Answering.** The primary goal of Multi-table question-answering evaluation is to measure the LLMs' proficiency in understanding complex queries, navigating through various tables, and generating correct and coherent answers. This evaluation is crucial for determining the effectiveness of LLMs in real-world applications, where they often need to interact with and extract information from multiple data sources simultaneously.

**Primary Key & Foreign Key Selection.** Unlike rows, which represent records in databases, columns represent attributes where the column header provides semantic meaning to the values. Moreover, the primary key, as well as the foreign key, is the important column feature of multi-table data. Hence a correct key selection interprets LLM's column-level understanding ability across multi-tables. Primary key & Foreign key selection is the percentage of correctly selected columns among all target columns in the evaluation set.

## 4 EXPERIMENTS

We design a series of experiments based on the MMQA benchmark, aiming to answer three questions: 1) How do LLMs perform on multi-table QA tasks compared to human performance? 2) What is the LLMs' performance on multi-table related tasks such as Primary Key Selection and Foreign Key Selection? 3) The average number of rows of MMQA is more than 1,000. How do LLMs perform on long tables?

### 4.1 EXPERIMENTAL SETTINGS

**Datasets.** Specifically, we divide our MMQA benchmark into two parts: 2-table (2591 samples, average of 1833.31 rows and 6.04 columns) and 3-table (721 samples average of 1369.01 rows and 4.78 columns) subsets.

**Models.** We employ the proprietary and open-source LLMs in our experiments and to enhance reproducibility, we set the temperature to 0.7 for proprietary models, and all the experiment results are the average scores of three experiment results. For proprietary models, we adopt GPT-4 (Achiam et al., 2023), GPT-3.5 (Ouyang et al., 2022), Gemini-pro (Team et al., 2023), and O1-preview. For open-source LLMs, we evaluate on TableLlama-7b (Zhang et al., 2023) and Mistral-7b (Jiang et al., 2023). The prompts of different evaluation tasks are shown in the Appendix B.

**Evaluation Metrics.** For the table retrieval task, we utilize precision, recall, and F1 scores to measure the performance of multi-table retrieval. For multi-table reasoning, LLMs were assessed using a combination of metrics, including Exact Match (EM) and Partial Match (PM) for multi-table QA, Rouge-1, Rouge-L (Lin, 2004), BLEU (Papineni et al., 2002) for Text-to-SQL task, and accuracy scores for Primary Key Selection (PKS) and Foreign Key Selection (FKS). Partial Match indicates the partial semantic match scores between LLMs' generated answers and gold answers. We utilize GPT-4-turbo as the answer evaluator to give the scores and the prompt is available in Appendix

Table 5: The main results of different baselines on the 2 table dataset. We divided our benchmark into a 3-table subset and a 2-table subset. We use * to denote the zero-shot setting and †to denote the one-shot setting. We here abbreviate the Primary Key Selection as "PKS", and the Foreign Key Selection as "FKS". PM indicates the partial semantic match of LLMs' generates evaluated with GPT-4-turbo with the prompt in Appendix B.

| Dataset | 2 table | | | | | | |
|---|---|---|---|---|---|---|---|
| Evaluation Methods | Table QA | | Text-to-SQL | | | PKS | FKS |
| Metrics | EM | PM | Rouge1 | RougeL | BLEU | Acc | Acc |
| *Open Source LLMs* | | | | | | | |
| TableLlama 7b* | 7.58±0.3 | 8.06±0.1 | 9.12 ±0.2 | 7.89 ±0.2 | 1.82±0.1 | 17.86±0.2 | 13.75±0.2 |
| TableLlama 7b† | 8.23±0.1 | 8.57±0.1 | 10.34 ±0.3 | 9.53 ±0.2 | 2.92±0.2 | 20.25±0.2 | 15.89±0.2 |
| Mistral-7b * | 5.36±0.1 | 5.89±0.1 | 7.25±0.2 | 6.36±0.1 | 1.79±0.1 | 14.15±0.1 | 13.98±0.2 |
| Mistral-7b † | 6.26±0.2 | 6.72±0.1 | 9.55±0.1 | 8.45±0.1 | 2.49±0.1 | 17.65±0.2 | 16.17±0.2 |
| LlaMA-2-13b * | 9.45±0.2 | 10.13±0.1 | 17.34±0.2 | 15.81±0.3 | 5.44±0.1 | 25.34±0.1 | 22.78±0.1 |
| LlaMA-2-13b † | 11.28±0.2 | 13.04±0.2 | 20.45±0.1 | 18.17±0.1 | 7.59±0.2 | 28.89±0.2 | 25.27±0.1 |
| *Proprietary LLMs* | | | | | | | |
| GPT-3.5* | 25.56±0.2 | 29.34±0.1 | 31.75±0.1 | 27.89±0.2 | 2.71±0.3 | 28.06±0.1 | 19.25±0.1 |
| GPT-3.5† | 26.79±0.1 | 29.78±0.1 | 33.96±0.2 | 29.74±0.1 | 4.82±0.2 | 38.75±0.1 | 28.83±0.1 |
| GPT-4* | 25.17±0.2 | 31.35±0.2 | 32.51 ±0.2 | 28.39±0.3 | 2.53±0.1 | 31.59 ±0.2 | 21.27 ±0.2 |
| GPT-4† | 28.88±0.1 | 34.57±0.1 | 39.64±0.2 | 35.07±0.2 | 5.77 ±0.2 | 42.78 ±0.2 | 26.88±0.1 |
| Gemini-pro* | 27.16±0.2 | 32.72±0.2 | 33.13±0.2 | 29.28±0.1 | 2.69±0.1 | 32.77±0.1 | 22.06±0.2 |
| Gemini-pro† | 28.58±0.2 | 33.89±0.1 | 35.26±0.1 | 30.15±0.2 | 5.34±0.2 | 44.19±0.2 | 28.38±0.2 |
| O1-preview* | 46.25±0.2 | 49.72±0.2 | 38.41±0.2 | 37.75±0.3 | 6.79±0.3 | 42.81±0.1 | 30.53±0.1 |
| O1-preview★ | 50.78±0.2 | 53.85±0.2 | 43.62±0.2 | 39.52±0.3 | 7.58±0.2 | 49.53±0.2 | 34.17±0.2 |
| **Human** | **89.8** | | **82.7** | | | **96.5** | **95.3** |

B. These metrics provide a comprehensive view of the models' performance, from their ability to generate accurate SQL queries (Text-to-SQL) to their capacity to understand and reason over tabular data.

## 4.2 MULTI-TABLE RETRIEVAL EVALUATION

Table 4 presents the results of a comprehensive experiment evaluating different baselines on a multi-table retrieval task on MMQA with different choices of K (K=2,5,10). Across different values of K, our MTR achieves SOTA performance compared to previous strong retrieval approaches (BM25, TF-IDF, DTR, and open-source LLMs). Notably, the BM25, TF-IDF, and Table Dense retrieval which are based on the small models show poor performance on retrieval of multiple interconnected tables. While SGPT-5.8B and TableLlama-7b could handle long structure input, the question-relevance score and table-relevance score in MTR comprehensively consider the intra- and inter-connections. The multi-table retrieval experiment results indicate a discernible variation in performance among the evaluated LLMs. Specifically, our MTR outperforms the open-source LLMs, SGPT, and TableLlama, and demonstrates superior precision, achieving a score of 72.3%, which suggests a high accuracy in identifying the most relevant tables for a given question. This performance is attributed to MTR's advanced question understanding and its ability to discern the intricacies of multi-table relations. In contrast, the MTR combined with TableLlama-7b while showing commendable recall with a score of 64.7%, lagged in precision, indicating a tendency to retrieve a broader set of tables that occasionally included less relevant ones. The F1 score, which harmonizes precision and recall, was highest for MTR at 68.3%, reflecting a balanced performance in both identifying relevant tables and minimizing false positives. These results underscore the importance of understanding table relationships in multi-table retrieval tasks, an area where MTR outperforms open-source counterparts. We also ablate the Question Decomposition (QD) module, and we found that QD plays a vital role in MTR. QD provides noticeable improvements over MTR on multi-table retrieval tasks, for instance, in Top-2 retrieval, precision is improved from 65.3 to 72.3 and recall is improved from 62.3 to 64.7 in the 2-table subset.

Table 6: The main results of LLMs on the 3-table subset.

| Dataset | 3 table | | | | | | |
|---|---|---|---|---|---|---|---|
| Evaluation Methods | Table QA | | Text-to-SQL | | | PKS | FKS |
| Metrics | EM | PM | Rouge1 | RougeL | BLEU | Acc | Acc |
| ***Open Source LLMs*** | | | | | | | |
| TableLlama-7b[*] | $7.42_{\pm0.1}$ | $8.12_{\pm0.1}$ | $8.96_{\pm0.2}$ | $7.58_{\pm0.3}$ | $1.77_{\pm0.1}$ | $16.17_{\pm0.2}$ | $11.57_{\pm0.2}$ |
| TableLlama-7b† | $7.82_{\pm0.1}$ | $8.38_{\pm0.1}$ | $9.36_{\pm0.1}$ | $7.92_{\pm0.3}$ | $2.15_{\pm0.1}$ | $18.05_{\pm0.1}$ | $13.57_{\pm0.2}$ |
| Mistral-7b [*] | $5.27_{\pm0.1}$ | $5.91_{\pm0.1}$ | $3.72_{\pm0.1}$ | $2.46_{\pm0.2}$ | $1.68_{\pm0.1}$ | $16.86_{\pm0.1}$ | $12.58_{\pm0.2}$ |
| Mistral-7b † | $5.88_{\pm0.2}$ | $6.26_{\pm0.1}$ | $4.33_{\pm0.2}$ | $3.08_{\pm0.2}$ | $2.58_{\pm0.1}$ | $18.24_{\pm0.1}$ | $15.06_{\pm0.2}$ |
| LlaMA-2-13b [*] | $8.62_{\pm0.1}$ | $9.24_{\pm0.2}$ | $14.22_{\pm0.2}$ | $12.75_{\pm0.1}$ | $4.79_{\pm0.1}$ | $21.27_{\pm0.1}$ | $20.09_{\pm0.1}$ |
| LlaMA-2-13b † | $9.65_{\pm0.1}$ | $11.74_{\pm0.1}$ | $18.66_{\pm0.2}$ | $15.73_{\pm0.2}$ | $6.29_{\pm0.2}$ | $24.37_{\pm0.2}$ | $22.58_{\pm0.1}$ |
| ***Proprietary LLMs*** | | | | | | | |
| GPT-3.5[*] | $20.66_{\pm0.2}$ | $24.29_{\pm0.1}$ | $31.48_{\pm0.3}$ | $27.39_{\pm0.3}$ | $2.67_{\pm0.1}$ | $27.36_{\pm0.2}$ | $18.18_{\pm0.3}$ |
| GPT-3.5† | $24.64_{\pm0.2}$ | $28.55_{\pm0.2}$ | $39.25_{\pm0.2}$ | $33.26_{\pm0.3}$ | $5.26_{\pm0.1}$ | $40.16_{\pm0.1}$ | $25.01_{\pm0.2}$ |
| GPT-4[*] | $27.16_{\pm0.2}$ | $31.48_{\pm0.2}$ | $33.13_{\pm0.2}$ | $29.28_{\pm0.1}$ | $2.69_{\pm0.1}$ | $32.77_{\pm0.1}$ | $22.06_{\pm0.2}$ |
| GPT-4† | $28.58_{\pm0.2}$ | $33.21_{\pm0.1}$ | $35.26_{\pm0.1}$ | $30.15_{\pm0.2}$ | $5.34_{\pm0.2}$ | $44.19_{\pm0.2}$ | $28.38_{\pm0.2}$ |
| Gemini-pro[*] | $24.25_{\pm0.1}$ | $28.59_{\pm0.1}$ | $29.78_{\pm0.2}$ | $26.17_{\pm0.2}$ | $3.02_{\pm0.1}$ | $30.31_{\pm0.1}$ | $21.92_{\pm0.2}$ |
| Gemini-pro† | $26.38_{\pm0.1}$ | $30.88_{\pm0.1}$ | $33.44_{\pm0.1}$ | $29.25_{\pm0.2}$ | $4.88_{\pm0.2}$ | $38.85_{\pm0.2}$ | $31.52_{\pm0.2}$ |
| O1-preview[*] | $42.37_{\pm0.1}$ | $45.97_{\pm0.2}$ | $36.29_{\pm0.2}$ | $35.73_{\pm0.3}$ | $5.28_{\pm0.1}$ | $41.89_{\pm0.2}$ | $32.32_{\pm0.2}$ |
| O1-preview† | $48.28_{\pm0.2}$ | $52.95_{\pm0.2}$ | $42.41_{\pm0.2}$ | $36.29_{\pm0.1}$ | $7.08_{\pm0.1}$ | $46.84_{\pm0.1}$ | $40.78_{\pm0.1}$ |
| Human | **92.3** | | **86.9** | | | **98.7** | **97.5** |

## 4.3    MULTI-TABLE REASONING EVALUATION

The main results of various LLMs are presented in Tables 5 and 6. Taking Table 5 as an example, O1-preview outperforms numerous multi-table tasks on our MMQA benchmark, demonstrating superior performance across complex reasoning scenarios. Particularly in the Table QA task (50.78 EM score), Primary Key Selection (49.53 accuracy), and Foreign Key Selection (34.17 accuracy), the O1-preview maintains a noticeable level of performance, significantly surpassing GPT-4, (28.88 EM score in Table QA, 42.78 accuracy in Primary Key Selection, and 26.88 in Foreign Key Selection). Despite these advancements, proprietary and open-source LLMs still lag significantly behind human performance (89.8, 82.7, 96.5 and 95.3) on multi-table comprehension and reasoning tasks. Nevertheless, certain advanced LLMs, especially table-related LLMs, demonstrate potential in these scenarios.

**Text-to-SQL.**    In the Text-to-SQL generation task, LLMs are assessed on their ability to translate natural language questions into accurate SQL queries. O1-preview with a one-shot setting emerged as a top performer, with Rouge1, RougeL, and BLEU scores of 43.62, 39.52, and 7.58 in Table 5, respectively. This high score can be attributed to O1-preiew's sophisticated language parsing and structured output generation capabilities, which are critical for understanding the semantic nuances of questions and mapping them onto the corresponding SQL syntax. GPT-4, while achieving a respectable BLEU score, showed lower Rouge-L and Rouge-1 scores, suggesting that although it could capture the overall structure of the SQL query, it struggled with the finer details of the query syntax. The Text-to-SQL results reveal that while LLMs are making strides in this area, there is still considerable room for improvement, particularly in generating queries that are not only syntactically correct but also semantically aligned with the original questions.

**Multi-table QA.**    The multi-table qa task evaluates the models' capacity to retrieve evidence, navigate across multiple tables, and finally extract correct answers. O1-preview with one-shot setting exhibited an impressive Exact Match score of 50.78 and 48.28 in Tables 5 and 6, showcasing its proficiency in comprehending complex, inter-table relationships to retrieve accurate answers. This performance is likely due to Model E's ability to effectively utilize foreign keys and primary keys to traverse between tables and identify the relevant data points. Conversely, GPT-4, despite a solid showing in other tasks, only managed an Exact Match score of 28.88, and 28.58, respectively, in Tables 5 and 6, suggesting that it encountered difficulties in integrating information from multiple tables to

form a coherent answer. The variance in performance across models highlights the complexity of multi-table reasoning and the need for LLMs to develop more sophisticated strategies for inter-table data integration.

**Primary Key Selection and Foreign Key Selection.** The accuracy of primary and foreign key selection is pivotal for establishing correct table relationships, which was tested in this task. O1-preview stood out with an accuracy score of 46.84% for Primary Key Selection and 40.78% for Foreign Key Selection in 6, indicating a robust understanding of table schemas and the ability to accurately identify critical columns that facilitate data linkage across tables. These high scores reflect O1-preview's advanced feature engineering capabilities and its nuanced grasp of database structures. On the other hand, GPT-4 while competent in other tasks, achieved lower accuracy scores of 42.78% and 26.88% for primary and foreign key selection, respectively. This performance discrepancy could be due to GPT-4's less refined ability to discern the significance of certain columns within the context of table relationships. The results from this task emphasize the importance of precise key identification for effective multi-table data processing, an aspect that remains a challenge for many LLMs and warrants further research and refinement.

**Impact of Table Length.** We randomly select data with average table lengths of 500, 600, 700, 800, 900, and 1,000, sampling 50 samples for each type of data to evaluate the performance of LLM under different length tables. Figures 4 and 5 present an insightful evaluation of the Multi-Table QA and Text-to-SQL Generation tasks' performance across various table lengths. Figure 4 reveals that there is an elbow point in the Multi-Table QA task. When the table length does not exceed 800 rows, the performance of LLMs decreases gently as the table length increases. But when the length of the table exceeds 800 rows, there is an elbow point where the performance of LLM rapidly drops to a significantly low level. While, in the Text-to-SQL Generation task, no matter how the length of the table changes, the performance of LLMs has maintained a slow decline rate. This may be because, in Text-to-SQL tasks, LLMs only focus on the table header rather than the table content itself.

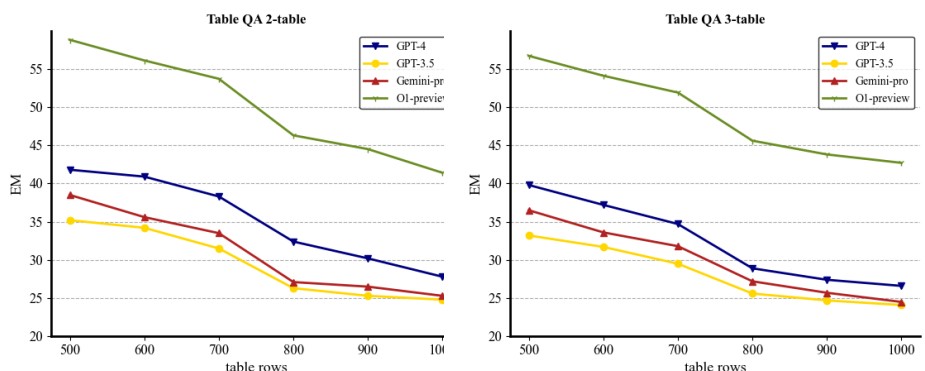

Figure 4: The evaluation of different lengths of input tables on multi-table QA task. We divided MMQA into 6 subsets: tables' lengths of 500, 600, 700, 800, 900, and 1000. All LLMs are evaluated on a zero-shot setting.

## 5    CONCLUSION

We introduce a new MMQA benchmark for assessing of LLMs' capabilities in handling multi-table tasks. The extensive experiments conducted reveal both the promise and limitations of current LLMs in navigating complex, interconnected data. Although existing strong LLMs such as GPT-4 and O1-preview showcase a strong performance on complex tasks, LLM still lacks the ability to comprehensively understand and reason over tables, especially in multi-table tasks, and lags significantly behind human performance in Multi-Table Question Answering tasks and Foreign Key Selection tasks. As the field progresses, the MMQA dataset and its associated challenges will undoubtedly serve as a critical catalyst for innovation, driving the development of LLMs that can more effectively be reasoning across multiple tables, tackling real-world data complexities.

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

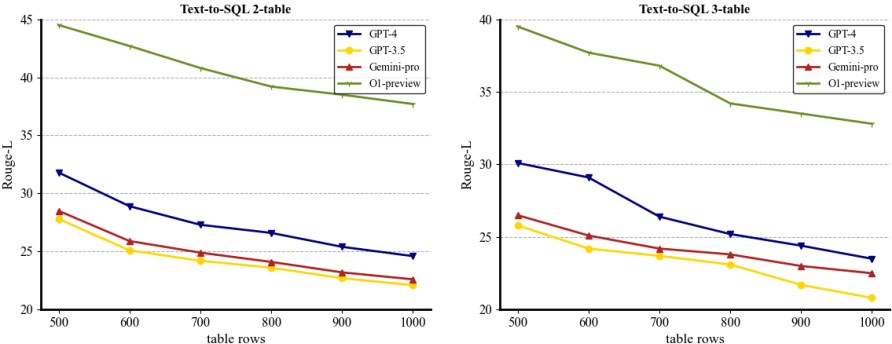

Figure 5: The evaluation of different lengths of input tables on Text-to-SQL Generation task. We divided MMQA into four subsets: tables' lengths around 500 (400-600), 700 (600-800), 900 (800-1000), and 1100 (1000-1200). All LLMs are evaluated on a zero-shot setting.

## A    REPRODUCIBILITY STATEMENT

To make the results and models reproducible and verifiable, we provide our full data annotation guideline, data link, implementation details, and prompts: We detail the process of data annotation in section 3.1 and the implementations are in Appendix C. All the prompts required to reproduce the results are illustrated in Appendix B.

## B    PROMPTS

When evaluating large language models, prompting is a brittle process wherein small modifications to the prompt can cause large variations in the model predictions, and therefore significant effort should be dedicated to designing a painstakingly crafted perfect prompt for the given task (Arora et al., 2022; Diao et al., 2023). In this study, We investigate the performance of zero-shot on our benchmark. To eliminate the randomness, we manually select one demonstration for each task, ensuring that all tasks are covered.

We give our designed input examples for three different tasks to help readers understand our implementation, as shown in Table 7, respectively.

## C    IMPLEMENTATION DETAILS

For proprietary models, we employ official APIs to interact with exclusive LLMs and prompts are well-defined. For open-source models, all experiments are conducted on 8 A100 GPUs. For fine-tuning single-table-retrieval models, we conduct supervised fine-tuning of TableLlama-7B and SGPT-5.8B on the single-table QA dataset. We set the initial learning rate at 2e-5 and conducted training over three epochs. Optimization is performed using the Adam optimizer, with a batch size of 4 and a maximum input sequence length of 4,096.

## D    CORRELATIONS OF PARTIAL MATCH BETWEEN GPT-4 AND HUMAN CHECK

We also check the person scores between GPT-4 Partial Match score and Human Check score. We randomly selected 100 data from MMQA (50 from the 2-table subset, 50 from the 3-table subset) and manually checked the partial match score. For answers generated by O1-preview, GPT-4 gives the 53 partial match score which indicates that 53 answers can be aligned to ground truth. Human check gives 59 partial score which indicates the 59 answers can be aligned to ground truth. We collect two lists with 100 elements, the element is "0" or "1". One list is the GPT-4 partial match score list and another is the Human Check partial match score list. We compute the Pearson Correlations between

Table 7: The prompt templates of table-related tasks. We here take 2 table data as an example. [WORDS] denotes the information we should provide.

---

***Prompts of Question Decomposition***

*Prompt* You are an expert at multi-hop question decomposition, you need to decompose the given multi-hop question [Question] based on the given example. Please only output the results without any other words in the JSON format of: {"Sub-questions": List}."
*[Question]* The given multi-hop question.
*[Example]* The given example of question and sub-questions.

---

***Prompts of Text-to-SQL***

*Prompt* You are an expert at text-to-SQL, you need to generate a SQL query based on the given multihop question [Question] and given two tables [TABLE1], [TABLE2]. Please only output the results without any other words in the JSON format of: {"SQL": String}. "
*[Question]* The given multi-hop question.
*[TABLE1]* The given table 1.
*[TABLE2]* The given table 2.

---

***Prompts of Multi-table QA***

*Prompt* "You are an expert at multi-table question answering, you need to extract answers based on the given multi-hop question [Question] and given two tables [TABLE1], and [TABLE2]. Please only output the results without any other words in the format of: {"Answers": List}. *[Question]* The given multi-hop question.
*[TABLE1]* The given table 1.
*[TABLE2]* The given table 2.

---

***Prompts of Foreign Key Selection***

*Prompt* "You are an expert at foreign key selection, you need to select foreign keys based on the given two tables [TABLE1], and [TABLE2]. Please only output the results without any other words in the JSON format of: {"foreign keys": List}.
*[TABLE1]* The given table 1.
*[TABLE2]* The given table 2.

---

***Prompts of Primary Key Selection***

*Prompt* "You are an expert at primary key selection, you need to select primary keys based on the given two tables [TABLE1], and [TABLE2]. Please only output the results without any other words in the JSON format of: {"primary keys": List}.
*[TABLE1]* The given table 1.
*[TABLE2]* The given table 2.

---

***Prompts of Partial Match Evalutaion***

*Prompt* You are an Answer evaluator, you need to measure the semantic similarity between [Generated Answer] and [Gold Answer], and give the score, 1 means equal, 0 means not. Some answers may have abbreviations or alias, for example, Lionel Messi is equal to Messi, Donald Trump is equal to Trump. Please only output the score 1 or 0 without any other words.
*[Generated Answer]* The LLM generated answer.
*[Gold Answer]* The ground truth.

---

the two lists. The results are illustrated in Table 8, and we find that the Human Check partial match score is highly correlated to the GPT-4 Partial Match score. For example, given the question "Find the number of male (sex is 'M') students who have some food type allergy." The answer is "10", while the generated answer is "ten". GPT-4 PM could treat "ten" as the correct answer. However, human partial match is better than GPT-4 PM, For example, the question is 'What are all the employee IDs and the names of the countries in which they work?" One of the answers is "CA", while the LLM-generated answer is "Canada". The GPT-4 PM assigned score is 0.

Table 8: Pearson Correlations between GPT-4 Partial Match Score and Human Check Partial Score.

| Model | EM | GPT-4 PM | Human Check PM | Pearson Correlation |
|---|---|---|---|---|
| O1-preview | 45.7 | 53 | 59 | 0.8852 |
| GPT-4 | 31.6 | 41 | 45 | 0.8273 |
| GPT-3.5 | 26.7 | 38 | 41 | 0.7784 |

# E  DIFFERENCE BETWEEN ORIGINAL QUESTIONS AND PARAPHRASED QUESTIONS

We randomly selected 100 questions (50 from the 2-table subset and 50 from the 3-table subset) that were paraphrased manually and sent the questions with corresponding tables into LLMs and evaluated Table QA tasks with EM score. The results are illustrated in Table 9. After paraphrasing, the table column-related information is reduced and the performance of LLMs also drops. For example, the original question is: "Show the name and number of employees for the departments managed by heads whose temporary acting value is 'Yes'?". The paraphrased question is: "What are the names and number of employees of the department heads who are acting now?" The column-related information such as "temporary" and "managed" are eliminated.

Table 9: Performance between original SQL query generated questions and paraphrased questions.

| Models | 2-Table | | 3-Table | |
|---|---|---|---|---|
| Settings | Original | Paraphrased | Original | Paraphrased |
| O1-preview | 43.5 | 40.7 | 39.8 | 34.4 |
| GPT-4 | 29.6 | 25.8 | 26.2 | 23.6 |
| GPT-3.5 | 26.3 | 23.1 | 24.5 | 21.9 |

# F  TEST SUITE ACCURACY EVALUATION FOR TEXT-TO-SQL

We utilize the ESM (Zhong et al., 2020) for evaluating LLMs' Text-to-SQL performance on our 2-table and 3-table subsets. Table 10 illustrates that LLMs although get a relatively high pass rate of the ESM score, there are still a large proportion of false positive SQL queries.

Table 10: The false positive/negative rate of the ESM metric.

| Models | 2-table | 3-table |
|---|---|---|
| GPT-4 | 11.5/24.7 | 13.6/27.8 |
| GPT-3.5 | 15.4/28.1 | 17.9/31.2 |
| O1-preview | 8.7/19.4 | 11.3/22.6 |

# G  LIMITATIONS

In this paper, we focus on the evaluation of LLMs' multi-table understanding reasoning ability on our annotated counterfactual MMQA dataset. Although LLMs show an obvious performance gap between humans, the evaluation methods remain improving, for example, the Exact Match is not sufficient for replying to the real results. Secondly, although LLMs could generate SQL queries of a relatively good quality, whether the generated SQL queries could be executed to get correct answers or not is still unknown.

