# OpenReview forum: "MMQA: Evaluating LLMs with Multi-Table Multi-Hop Complex Questions"
_ICLR.cc/2025/Conference — ICLR 2025 Oral_

### Official Review · Reviewer_VktP · 2024-11-04

**Soundness:** 4
**Presentation:** 3
**Contribution:** 3
**Rating:** 8
**Confidence:** 4

**Summary:**

This paper introduces MMQA, a novel multi-table multi-hop question-answering dataset. The authors randomly sampled from the Spider dataset and prompted LLMs to generate natural language questions. The synthesized dataset then went through human experts' manual verification. Compared to existing table question-answering datasets, MMQA features a more diverse task collection, longer reasoning chains, and evaluations on different granularities. Extensive experiments run on MMQA indicate that there is still a large performance gap between the current SOTA LLMs and humans. The authors also propose a strong baseline, MTR, for the multi-table retrieval task. Additional insights and analysis on the evaluation results are also provided.

**Strengths:**

## **1. The task is novel and important**

The task of multi-table question answering is novel and important because it is very common in real product environments to consider multiple tables together, yet previous table question answering datasets have mainly focused on single-table scenarios. MMQA is able to close this gap. From this perspective, I believe the paper is well motivated and the problem is realistic and important.

## **2. The quality of the dataset is high**

The dataset contains 3,312 expert-verified tables with 7.61 average reasoning steps and a diversity of task types. The experiment results show a large gap where current SOTA LLMs such as OpenAI o1-preview only achieve around 50% success on MMQA evaluation metrics while humans achieved 80-90%.

## **3. A strong baseline is proposed**

The authors also proposed a strong baseline, MTR, that significantly (around 10-15% better on retrieval metrics) outperformed existing LLMs on the multi-table retrieving task. This model can serve as a good reference for future models on MMQA.

## **4. The paper is in general well written**
The paper is in general clear and easy to follow. The logic flow is straightforward and clearly presented. The authors also submitted supplementary dataset samples that give a sense of what the dataset will look like, which is helpful.

**Weaknesses:**

## **1. Diversity and naturalness of queries in the dataset**

The dataset is constructed by sampling from 45 SQL templates in the Spider dataset and converting them into natural language using GPT-4-turbo. This approach inherently limits the diversity of query types. Additionally, due to the synthetic nature of the dataset, its distribution does not fully align with that of real-world user queries.

## **2. Baseline models tested are relatively weak**

The current baselines are mostly general-purpose LLMs except TableLlama, which is a smaller 7B model. I would recommend adding more recent LLM-based table question answering agentic systems as baselines, such as the ones on top of the Spider leaderboard: https://yale-lily.github.io/spider

**Questions:**

## **1. What kinds of annotation disagreement occurred?**

In 3.1 the authors described the annotation quality verification process. I was wondering what kinds of disagreement patterns were observed. Were the annotators disagreeing with each other because they made mistakes themselves all the time? Was there any case that was due to query ambiguity or GPT-4-turbo's incorrect NL translation? How did you manage those situations, if any?

## **2. Did you notice any bias, information loss, or noises introduced by GPT-4-turbo as the query translator?**

One concern about GPT-4-turbo translating the template-based queries is that GPT-4-turbo might introduce bias (e.g., some wording patterns get overrepresented), information loss (e.g., the semantics in the SQL might not be fully faithfully translated into NL), or noises (e.g., hallucinated or ignored columns/entities/tables/etc). Did you observe any of such risks and how did you handle them? Would it make sense to also include the caveats and potential issues with the approach in e.g. appendix?

---

> ### Author Response · Authors · 2024-11-21
> **Response to reviewer VktP**
>
> **Response weakness 1**:
>
> Thank you for the insightful comments, we selected 100 questions (50 from the 2-table subset and 50 from the 3-table subset) that are paraphrased manually and sent the questions with corresponding tables into LLMs and evaluated Table QA tasks with EM score:
>
> | Models  | 2-Table   | 2-Table  | 3-Table | 3-Table |
> |-|:---:|:---:|:---:|:---:|
> | Settings  | Original   | Paraphrased  | Original | Paraphrased |
> | O1-preview     | 43.5     | 40.7              | 39.8                                   | 34.4                           |
> | GPT-4       | 29.6    | 25.8            | 26.2                                   | 23.6                           |
> | GPT-3.5          | 26.3     | 23.1              | 24.5                                   | 21.9                           |
> ***Table 1. Evaluated Table QA tasks with EM score.***
>
> **For Example:**
> - The original Question: “Show the name and number of employees for the departments managed by heads whose temporary acting value is 'Yes'?”
> - Paraphrased Question: “What are the names and number of employees of the department heads who are acting now?”
>
> The column names of the corresponding tables are: ["Department_ID", "Department_Name", "Num_Employees", "Head_ID", "Head_Name", "Temporary_Acting"]
>
> In paraphrased questions, there are still some keywords, such as employees and department, that can not be removed.
> After paraphrasing, the performance of LLMs drops.
>
> We updated the above-mentioned experimental results and analysis in Appendix E.
>
> **Response weakness 2**:
>
> We utilized the DAIL-SQL + GPT-4 + Self-Consistency [1], DIN-SQL + GPT-4 [2], and LlaMA 2 13B as the baselines on text-to-SQL task:
>
> | Models  | 2-Table   | 2-Table  | 2-Table | 3-Table | 3-Table | 3-Table |
> |-|:---:|:---:|:---:|:---:|:---:|:---:|
> | Metrics| Rouge-1   | Rouge-L  | BLEU | Rouge-1 | Rouge-L | BLEU|
> | DAIL-SQL     | 45.14     | 42.30              | 9.67     | 41.37 | 36.52 | 8.33 |
> | DIN-SQ      | 43.62    | 39.58           | 8.09  | 38.58 | 35.69 | 7.69 |
> | LlaMA 2 13B          | 17.34    | 15.81  | 5.44 | 14.22 | 12.75 | 4.79 |
> ***Table 2. Comparison of baselines on the text-to-SQL task.***
>
> Since  DAIL-SQL + GPT-4 + Self-Consistency and DIN-SQL + GPT-4 are text-to-SQL tasks, which is not applicable to other subtasks of MMQA.
> We added the LlaMA-2-13B results of all subtasks in Tables 5 and 6.
>
> ***References***
> [1] Text-to-SQL Empowered by Large Language Models: A Benchmark Evaluation, VLDB, 2023.
> [2] DIN-SQL: Decomposed In-Context Learning of Text-to-SQL with Self-Correction, NeurIPS, 2023.
>
> **Response question 1**:  Thank you for the insightful comments. In the annotation process, two annotators were asked to annotate primary keys, foreign keys, and answers of the multi-hop questions. For primary key and foreign key annotation, the disagreement often happens on the intermediate tables. For example, in table Management in Figure 1, both Department ID and Head ID are primary keys and foreign keys.
>
> Secondly, in the answer annotation, some questions such as "How many departments are led by heads who are not mentioned?" The answer is a certain number of departments. However, our table usually contains more than 100 rows, so annotators may make mistakes in counting numbers. A third expert is required to check the number.
>
> **Response question 2**:
>
> Thank you for the insightful comments. Yes, the overrepresented situation exists.
> For example:
> - The SQL query is: "SELECT DISTINCT T1.age FROM management AS T2 JOIN head AS T1 ON T1.head_id = T2.head_id WHERE T2.temporary_acting = 'Yes'"
> - The GPT-4 generated question is: "What are the distinct ages of individuals in the 'head' table who are associated with a 'management' entry where the 'temporary_acting' status is 'Yes'?"
>
> - Where the final question is: "What are the distinct ages of the heads who are acting?"
> Some overrepresented questions are paraphrased by human annotators. Some wrong translations are discarded.

---

> > ### Comment · Reviewer_VktP · 2024-11-23
> >
> > Thank you for your response, I think that addressed my questions.

---

### Official Review · Reviewer_PPNX · 2024-11-04

**Soundness:** 3
**Presentation:** 3
**Contribution:** 3
**Rating:** 8
**Confidence:** 5

**Summary:**

This paper addresses the task of information retrieval and question answering from multiple tables, focusing on questions that require information from multiple tables and reasoning. For this task, the paper introduces the _Multi-table and Multi-hop Question Answering (MMQA) dataset_, along with several subtasks that multi-table QA systems might need to solve: Table retrieval, Table QA, Text-to-SQL, Primary key selection, and Foreign key selection.

This dataset is constructed by template-based synthesis of SQL queries from the tables in the Spider dataset, and converting them into natural language questions using GPT-4-Turbo, in addition to quality checks conducted by experts.

The paper also introduces _MTR_, a new method for the multi-table retrieval subtask. MTR uses question decomposition and both question-table relevance (using single-table retrievers from prior work) and table-table relevance, to tackle this subtask.

Finally, the paper benchmarks several open LLMs (TableLLaMA-7B and Mistral-7B) and proprietary LLMs (GPT-3.5, GPT-4, Gemini-pro, and O1) on the MMQA dataset and conducts ablation studies to understand the effects of table length on the performance of various systems. These experiments show that there is a significant gap between the performance of LLMs and human experts on this dataset and task.

**Strengths:**

This paper identifies a task where large language models (LLMs) are significantly behind humans, and open models lag behind closed commercial models (like O1). This is quite valuable for the research community to continue developing better LLMs.

MMQA has a comprehensive set of question categories, including "list/count all answers," which are often overlooked in TableQA datasets.

The experiments are thorough, and include a good mixture of open and closed LLMs. I imagine future research will be able to easily compare their systems against these baselines.

**Weaknesses:**

## 1. The MMQA Dataset is Synthetic
To construct MMQA, this paper generates SQL using templates and then converts them to natural language using an LLM. There are two potential downsides to this approach. One is that synthetic evaluation and test sets pose the risk of overestimating the performance of models [4]. This may occur due to the lower diversity of natural language questions or the fact that, since they are generated from SQL, they may be too close to SQL or otherwise "unnatural" compared to questions that users of such systems may ask.

For example, the question in Figure 1, "What are the distinct creation years of the departments managed by a secretary born in state Alabama?" uses the terminology from the tables ("creation," "department," "state") and the SQL keyword "distinct." In a more realistic setting, a user might instead ask, "When were the departments managed by an Alabamian secretary established?". Given that the user is most likely not aware of column names in the database. Tackling this setting involves additional challenges like mapping the terms users use into column names and dealing with ambiguities.


## 2. Evaluation Metrics Need More Thought
Using ROUGE and BLEU for SQL is not really appropriate. To compare SQL queries directly, exact match is often used. Execution accuracy is another commonly used evaluation metric in the literature. It involves executing the resulting SQL against the database and comparing its answer against the answer from the gold SQL. Please note that the original Spider dataset paper uses these two metrics.
Even more suitable metrics are discussed in [3] and [2]. For example, [2] proposes a generalization of execution accuracy, which could be useful for evaluating "list" and "count" questions in MMQA.

Using Exact Match and Partial Match for QA underestimates the performance of models. This is especially noticeable when evaluating LLMs (as this paper does). Also, I wonder how these metrics handle "list" answers, given that they are quite sensitive to order. Or if a system is off by one when answering a "count" question because it missed one of N rows, shouldn't it get a partial score? See [5] for an example of how using an LLM judge to compare a model's predictions against the gold answer can help alleviate this problem.

In addition, it might make sense to add a unified text-to-SQL and TableQA subtask. This means that a system proposed in future work can answer the question in any way it chooses, either by generating and executing an SQL query or by directly reading the tables, and the final answer is evaluated against the gold answer. This would have the additional benefit that text-to-SQL methods and TableQA methods can use this dataset and compare their work across the two lines of research.

## Other minor issues
- In Algorithm 1, it should also be specified that a single-table retriever is used, in the same way it mentions that GPT-4-Turbo is used. Additionally, “outputs” are not sub-questions but tables.

## Suggestions
- Please consider adding a setting for directly benchmarking text-to-SQL without first retrieving tables. It is conceivable that in the near future, as long-context LLMs improve, they can generate SQL directly given the entire schema as input.

- To improve clarity, there could be a section dedicated to describing the various subtasks. For example, Section 3.3 could be renamed and dedicated to “subtasks.”

- MTR uses both query-table relevance and table-table relevance. I’m curious about the impact of each on the final quality of the retrieval system. An ablation study here would be informative.

- [1] converts BIRD and Spider text-to-SQL datasets to the open-domain format, where table retrieval is needed. What is the difference between that and MMQA? This could be added to Table 2 and Figure 3.

- There are many other Question Decomposition papers e.g. [6]; using one of them in MTR would be interesting.


**References:**
1. Is Table Retrieval a Solved Problem? Exploring Join-Aware Multi-Table Retrieval

2. SPINACH: SPARQL-Based Information Navigation for Challenging Real-World Questions

3. Semantic Evaluation for Text-to-SQL with Distilled Test Suites

4. AutoQA: From Databases To QA Semantic Parsers With Only Synthetic Training Data

5. Evaluating Open-Domain Question Answering in the Era of Large Language Models

6. Break It Down: A Question Understanding Benchmark

**Questions:**

1. In Table 1, what is the unit of "question length"? Is it measured in characters?

1. Line 199: How do you measure the number of reasoning steps?

1. What is the input for the Table QA, Text-to-SQL, "Primary Keys Selection," and "Foreign Key Selection" tasks? Are the gold tables used as input?

1. In Table 4, what is the difference between, for example, "TableLlama-7b" and "MTR (TableLlama-7b)"? Is it just the question decomposition part done by GPT-4-Turbo, or is their training data also different?

---

> ### Author Response · Authors · 2024-11-21
> **Response to Reviewer PPNX**
>
> **Response weakness 1:**
>
> Thanks for your insightful question. We randomly selected 100 questions (50 from the 2-table subset and 50 from the 3-table subset) that are paraphrased manually and sent the questions with corresponding tables into LLMs and evaluated Table QA tasks with EM score:
>
> | Models  | 2-Table   | 2-Table  | 3-Table | 3-Table |
> |-|:---:|:---:|:---:|:---:|
> | Settings  | Original   | Paraphrased  | Original | Paraphrased |
> | O1-preview     | 43.5     | 40.7              | 39.8                                   | 34.4                           |
> | GPT-4       | 29.6    | 25.8            | 26.2                                   | 23.6                           |
> | GPT-3.5          | 26.3     | 23.1              | 24.5                                   | 21.9                           |
>
> ***Table 1. Evaluated Table QA tasks with EM score.***
>
> ***For Example:***
> - The original Question: "Show the name and number of employees for the departments managed by heads whose temporary acting value is 'Yes'?"
>
> - Paraphrased Question: What are the names and number of employees of the department heads who are acting now?
>
> The column names of the corresponding tables are: ["Department_ID", "Department_Name", "Num_Employees", "Head_ID", "Head_Name", "Temporary_Acting"]
>
> In paraphrased questions, there are still some keywords, such as employees and department, that can not be removed.
> After paraphrasing, the performance of LLMs drops.
>
> We updated the above-mentioned experimental results and analysis in Appendix E of the current paper.
>
> **Response weakness 2**:
>
> We utilized the ESM(exact set match) metric of papers [3] as the metric for evaluating LLMs’ text-to_SQL performance on 2-table and 3-table subsets.
>
> | Models  | 2-Table   |  3-Table |
> |-|:---:|:---:|
> | GPT-4       | 11.5/24.7    | 13.6/27.8            |
> | GPT-3.5          | 15.4/28.1     |  17.9/31.2              |
> | O1-preview     | 8.7/19.4    | 11.3/22.6             |
>
> ***Table 2. Evaluating LLMs’ text-to_SQL performance. The false positive/negative rate of the ESM metric.***
>
> We updated the experiments in Appendix F of the current paper.
>
> **Response minor issues**:
>
> Thank you for the suggestions, we updated the description of the algorithm and equation （1）in the current paper, words in red.
>
> **Response-Suggestion 1**:
>
> We appreciate the valuable comments and sorry for the misunderstanding. In our experiments , we directly input the gold tables as the schema into LLMs and show the performance on the gold tables. It is because the Table retrieval module does perform well on the table retrieval task. The error propagation occurs.
>
> **Response Suggstion 2**:
>
> Thanks for your suggestion, we renamed Section 3.3 and dedicated it to “subtasks”.
>
> **Response Suggestion 3**:
> We have conducted the ablation study on the 2-table datasets. We list the results of the Top 2 Table retrieval task.
>
> | Models  | P   |  R | F1|
> |-|:---:|:---:|:---:|
> | MTR       | 72.3    | 64.7            | 68.3|
> | MTR w/o QT          | 27.9     |  21.3              | 24.2|
> | MTR w/o TT     | 57.8    | 45.8            | 51.1|
>
> ***Table 3. Ablation study on the 2-table datasets. Abbreviate the question-table relevance score as "QT" and the table-relevance score as "TT".***
> The question-table relevance score plays a more vital role in the Table retrieval task.
> We will add the full experiments including Top 5 retrieval and Top 10 retrieval in the final version of our paper.
>
> **Response suggestion 4**:
> The main differences between MMQA and Ref.[1] lies in 1) MMQA offers a more comprehensive benchmark on several tasks. [1] only focus on the Table retrieval task; 2) Our MTR method iteratively retrieves tables based on decomposed sub-questions. While [1] joint considers the question-table relevance and table relevance score and retrieves all tables. Finally, rank table scores. We added the statistics of Spider and BIRD in Table 2.
>
> **Response suggestion 5**:
> Thank you very much for your recommendation, we would add Ref.[6] in our paper.  We also utilized the DecompRC [1] and QDMR [2] as question decomposition baselines for MTR and compared the performance on the Multi-table Retrieval task:
>
> | Models  | P   |  R | F1|
> |-|:---:|:---:|:---:|
> | MTR (GPT-4)       | 72.3    | 64.7            | 68.3|
> | MTR (DecompRC)          | 64.2     |  57.8              | 60.9|
> | MTR (QDMR)    | 55.3    | 47.9            | 51.3|
> ***Table 4. DecompRC [1] and QDMR [2] as question decomposition baselines for MTR.***

---

> > ### Author Response · Authors · 2024-11-21
> > **Response to Reviewer PPNX**
> >
> > An example of decomposition results of MMQA question:
> > - Original Question: "Show distinct names of singers that have songs with sales of more than 300000."
> > - GPT-4 generated sub-questions:
> >     - Which songs have sales greater than 300,000?
> >     - What is the distinct name of the singers?
> > - DecompRC generated sub-questions:
> >     - Which singers have songs with sales of more than 300000?
> >     - What is the distinct name of [Answer of Q1]?
> > - QDMR generated sub-questions:
> >     - Return songs with sales of more than 3000?
> >     - Return singers of the #1
> >     - Return  distinct names of the #2
> >
> > We found that GPT-4 could generate more fluent, complete sub-questions for table retrieval.
> > ***Reference***
> > [1] Multi-hop Reading Comprehension through Question Decomposition and Rescoring, ACL, 2019.
> > [2] Break It Down: A Question Understanding Benchmark, TACL, 2020.
> >
> > **Questions-1**: In Table 1, what is the unit of "question length"? Is it measured in characters?
> >
> > **Response-Q1**: Yes, it measures the number of characters.
> >
> > **Questions-2**: Line 199: How do you measure the number of reasoning steps?
> >
> > **Response-Q2**: We count the number of reasoning times from question to final answer. In Figure 1, the number of reasoning steps is 6 as question->Head table->Head ID->Management table->Department ID->Department table->answer. The number of reasoning steps is 6.
> >
> >
> > **Questions-3**: What is the input for the Table QA, Text-to-SQL, "Primary Keys Selection," and "Foreign Key Selection" tasks? Are the gold tables used as input?
> >
> > **Response-Q3**: Yes, the gold tables. This is because when we use tables retrieved by MTR as inputs, error propagation occurs, which leads to worse performance in the Table QA, Text-to-SQL, "Primary Keys Selection," and "Foreign Key Selection" tasks.
> >
> > **Questions-4**: In Table 4, what is the difference between, for example, "TableLlama-7b" and "MTR (TableLlama-7b)"? Is it just the question decomposition part done by GPT-4-Turbo, or is their training data also different?
> >
> > **Response-Q4**: MTR (TableLlama-7B) leverages the TableLlama-7B model as a single-table retrieval module. We fine-tuned TableLlama-7B on publicly available single-table QA datasets to enhance its performance in this specific task.
> >
> > **Response add reference**:
> > We updated the references [1,2,4] in the current paper, where [1] is referred to in section 3.2, compared with our MTR. [2] is updated in Table 2, compared with our MMQA dataset. For [3], which is useful evaluation metrics on our subtasks, we added experiments in appendix F of our current paper.
> > [6] proposes a question decomposition dataset and a question decomposition method. We compared the effectiveness between QDMR and LLM question decomposition in the table in the previous text.

---

> > > ### Comment · Reviewer_PPNX · 2024-11-24
> > >
> > > Thank you for providing the additional information. This addressed some of my concerns.
> > > I recommend that -in future version of this paper- please move the ESM metric results to the main table.

---

### Official Review · Reviewer_P67K · 2024-11-04

**Soundness:** 4
**Presentation:** 4
**Contribution:** 4
**Rating:** 8
**Confidence:** 4

**Summary:**

This paper introduces a new benchmark for multi-table multi-hop reasoning using question answering. This benchmark evaluates models on 5 aspects: multi-table retrieval, text-to-sql generation, multi-table QA, Foreign Key selection, and Primary Key Selection. They also provide 2 splits: one for questions where reasoning is needed across 2 tables, and another one across 3 tables (and hence more challenging). This benchmark is novel because prior works focus on single-table QA, while the proposed one focuses on multi-table, which is also a more realistic setup since it is common to perform unions and joins across tables in real-world scenarios. They conduct evaluations on multiple proprietary LLMs and open-weights LLMs and compare them with human and baselines and multiple retrieval methods. The paper aims to answer the following questions:

How well are LLMs at solving multi-table QA tasks compared to humans? -> there is still a large gap to close
How well can LLMs solve structured reasoning tasks such as Primary/Foreign Key Selection? -> only o1 achieves an accuracy > 40, which shows LLMs struggle on this subtask, which is key for proper multi-table reasoning.
How does the performance of LLMs decrease when the table sizes increases? -> when the table is too long, performance rapidly decreases indicating there is a (length, #rows) threshold where LLMs cannot understand tables well.

**Strengths:**

* Very complete benchmark for evaluating table reasoning in LLMs.
* The benchmark includes different subsets for different difficulty levels, it also includes multiple subtasks, which can be useful to evaluate intermediate steps needed to solve multi-table QA.
* Analysis of multiple open-weights and close-weights LLMs.
* Includes short and longs tables, which can be useful for analysis of LLMs’ long-context understanding.
* More generally, this benchmark can be very useful for future research on table reasoning.

**Weaknesses:**

* The paper uses a partial match evaluation that depends on GPT4 (which can be fine), but they don’t provide and evaluation of the quality of this metric.

**Questions:**

* How good is the partial match evaluation? Does this metric correlate to a human partial match evaluation?

---

> ### Author Response · Authors · 2024-11-21
> **Response to Reviewer P67K**
>
> Thanks for the thoughtful and insightful comments, we will try to answer your questions as thoroughly as possible.
> Here, we randomly selected 100 candidate answers and ground truth from our MMQA and evaluated the GPT-4 partial match on the Table QA task.
>
>
> | Model  | EM    | GPT-4 PM  | Human Check PM | Pearson Correlation |
> |-|:---:|:---:|:---:|:---:|
> | O1-preview     | 45.7     | 53              | 59                                   | 0.8852                           |
> | GPT-4       | 31.6    | 41            | 45                                   | 0.8273                           |
> | GPT-3.5          | 26.7     | 38              | 41                                   | 0.7784                           |
> ***Table 1. Partial match evaluation of GPT-4 and humans.***
>
> For answers generated by O1-preview,  GPT-4 gives the 53 partial match score which indicates that 53 answers can be aligned to ground truth. Human check gives the 59 partial match score which indicates the 59 answers can be aligned to ground truth.
> For example, given the question ''Find the number of male (sex is 'M') students who have some food type allergy''. The answer is ''10'', while the generated answer is ''ten''. GPT-4 PM could treat ''ten'' as the correct answer.
>
> However, the human partial match is better than GPT-4 PM. For example, the question is ''What are all the employee ids and the names of the countries in which they work?'' One of the answers is ''CA'', while the generated answer of LLM is ''Canada''. The GPT-4 PM assigned score is 0.
>
> We updated the analysis in Appendix D of the current paper:  ''CORRELATIONS OF PARTIAL MATCH BETWEEN GPT-4 AND HUMAN CHECK''. of our current version.

---

> > ### Comment · Reviewer_P67K · 2024-11-25
> >
> > Thank you for your additional experiment. This addresses my question and shows that the partial match evaluation is good.

---

### Public Comment · ~Vinay_Kumar_Verma1 · 2025-06-09
**Data and Code link is not working**

Hi authors,
I find the paper is very interesting, thanks for the great work.

The link given in the paper is not working.
Can you provide the updated link to get the dataset and baselines code?

Thanks

---

> ### Public Comment · ~Yue_Zhang33 · 2025-06-15
> **Hi I found they released the data at github page**
>
> Hi,
>
> I found this issue page has the dataset
>
> https://github.com/WuJian1995/MMQA/issues/2
>
> Hope it works for you
>
> Best

---

> > ### Public Comment · ~Saptarshi_Sengupta1 · 2026-04-15
> >
> > This link also doesn't exist. If you do a Google search, the repo that comes up also doesn't work...

---

> > > ### Public Comment · ~Jian_Wu8 · 2026-04-15
> > >
> > > https://drive.google.com/drive/folders/1XQ9djKSK4yjxLWAmHMzsyPAKMqdCIpXo?usp=drive_link please try this

---

### Meta-Review · Area_Chair_Y4cD · 2024-12-20

**Metareview:**

This paper introduces the Multi-table and Multi-hop Question Answering (MMQA) dataset, a pioneering benchmark designed to assess the capabilities of large language models (LLMs) in handling complex multi-table data scenarios, which demand advanced understanding and reasoning across connected tables. The authors propose a comprehensive evaluation framework and a novel multi-table retrieval method that demonstrates state-of-the-art performance on the MMQA dataset. The reviewers are unanimous in their strong support for the paper, citing its significant contributions to advancing LLMs' abilities in real-world applications, hence I also recommend accepting the paper.

**Additional Comments On Reviewer Discussion:**

Nil

---

### Decision · Program_Chairs · 2025-01-22

Accept (Oral)